# The Landscape of Targeted Therapies in TNBC

**DOI:** 10.3390/cancers12040916

**Published:** 2020-04-08

**Authors:** Elena Vagia, Devalingam Mahalingam, Massimo Cristofanilli

**Affiliations:** Division of Hematology Oncology, Department of Medicine, Northwestern University Feinberg School of Medicine, Chicago, IL 60611, USA; mahalingam@nm.org (D.M.); massimo.cristofanilli@nm.org (M.C.)

**Keywords:** triple negative breast cancer, molecular profiling, DNA damage repair, targeted treatment, personalized medicine

## Abstract

Triple negative breast cancer (TNBC) constitutes the most aggressive molecular subtype among breast tumors. Despite progress on the underlying tumor biology, clinical outcomes for TNBC unfortunately remain poor. The median overall survival for patients with metastatic TNBC is approximately eighteen months. Chemotherapy is the mainstay of treatment while there is a growing body of evidence that targeted therapies may be on the horizon with poly-ADP-ribose polymerase (PARP) and immune check-point inhibitors already established in the treatment paradigm of TNBC. A large number of novel therapeutic agents are being evaluated for their efficacy in TNBC. As novel therapeutics are now incorporated into clinical practice, it is clear that tumor heterogeneity and clonal evolution can result to de novo or acquired treatment resistance. As precision medicine and next generation sequencing is part of cancer diagnostics, tailored treatment approaches based on the expression of molecular markers are currently being implemented in clinical practice and clinical trial design. The scope of this review is to highlight the most relevant current knowledge regarding underlying molecular profile of TNBC and its potential application in clinical practice.

## 1. Introduction

Breast cancer (BC), according to the American Cancer Society statistics 2019, is considered the most frequent cancer diagnosed and remains the second leading cause of cancer-related death, after lung cancer, among women in the United States [1]. BC constitutes a very heterogeneous disease. Three major tumor subtypes, hormone receptor (HR) positive, HER2-enriched and triple negative, are defined based on the immunohistochemical expression of estrogen and progesterone receptors and HER2 amplification or overexpression. This histopathological differentiation was the first crucial step for the implementation of more precise treatments for BC validating tumor heterogeneity and the need to assess and treat these subtypes as utterly distinct entities. Forthcoming molecular evolution further developed this concept, extending the diversity inside the tumor subtypes. It became evident that tumor molecular signature is an extremely important aspect of tumor evaluation and should be taken in consideration to guide treatment decisions. 

Triple negative breast cancer (TNBC) lacks the expression of hormone receptors (ER/PR) and demonstrates the overexpression of human epidermal growth factor (HER2). It accounts for 10–15% of all breast cancers [2]. TNBC usually appears in the form of high-grade invasive ductal carcinoma and has a higher rate of early recurrences, often with distant metastases and is associated with poorer prognosis compared to other BC subtypes. It occurs at a higher frequency in younger premenopausal women and is more prevalent among African-American descendants [3]. The evolution of the endocrine therapy for hormone receptor positive disease and anti-HER2 targeted treatment for HER2 amplified tumors have resulted in significant improvement of prognoses for these patients [4,5]. Despite the progress made to elucidate tumor biology, clinical outcomes for TNBC unfortunately remain unsatisfactory. The median overall survival for metastatic patients is approximately eighteen months, much lower compared to the HR positive and HER2 enriched disease where survival may exceed five years [6]. This fact emphasizes the compelling need to develop more effective treatments for TNBC patients. The scope of this review is to highlight the most relevant knowledge about underlying molecular profile of TNBC and its potential application in clinical practice.

## 2. Molecular Profile of TNBC

### 2.1. Intrinsic Subtypes 

Over the past decade, efforts have been made to elucidate the molecular profile of TNBC translating in valuable information on the heterogeneity and biological complexity of this subtype at molecular level. that accounts for the aggressive behavior of TNBC allowing researchers to modify treatment approaches. Below, we will review the most relevant data regarding molecular classification of TN tumors, which constitutes the foundation for therapeutic development. 

Lehman and colleagues first reported further classification of TNBC based on the tumor gene expressing profile. Analyzing microarray data, they identified six molecular subtypes characterized not only by distinct patterns of gene expression signature but also very diverse clinical behavior [7]. 

Basal-like subtypes (BL1 and BL2) constitute approximately 70% of TN tumors. BL1 expresses genes involved in the cell-cycle regulation, cell proliferation and DNA damage response while BL2 has high expression in genes involved in the cell cycle, cell division and growth factor signaling [8,9]. BL1 tumors are almost exclusively of ductal histology, high grade and achieve higher pathological complete response (pCR) rate when given neoadjuvant chemotherapy. Explicitly, BL2 subtype tumors are less likely to achieve pCR and has higher risk of recurrence compare to BL1. For aggressive tumors such as TNBC, pCR is accepted as a favorable prognostic marker associated with long-term survival benefit. 

Mesenchymal (M) and Mesenchymal stem-like (MSL) subtypes demonstrate high expression of genes involved in epithelial-mesenchymal transition and growth factor signaling (*EGFR, PDGFR, PI3K/mTOR, Src*). Tumors under the M category are less sensitive to chemotherapy compared to BL1 and preferentially metastasize to lungs. Metaplastic carcinomas are mostly M subtype. 

Immunomodulatory subtype (IM) expresses genes involved in antigen presentation and immune processing *NFKB, TNF, JAK*, and cytokine signaling. This subtype demonstrates a better prognosis among TNBCs. Later data showed that differences in gene overexpression of IM and MSL subtypes are predominantly derived from the tumor microenvironment such as infiltrating immune cells and tumor-associated mesenchymal tissue respectively. Interestingly and in concordance with the above results, these genes are not expressed in cell lines when performing in vitro experiments, where the microenvironment is absent. Tumor microenvironment clearly has a very important role on the tumor growth, treatment response and resistance and needs to be investigated in further details. In view of immunotherapy evolution, IM subtype was refined by the authors and represents an immune modulation of the tumor rather than a distinct subtype [10]. The high expression of immune-regulator CTLA4, PD1, and PD-L1 in TNBC generated from lymphocyte infiltration of the tumor, is probably associated with response to immune checkpoint inhibitors regardless the molecular subtype [11]. Furthermore, multiple studies are suggesting that the presence of tumor-infiltrating lymphocytes (TILs) might be predictive for better survival outcomes. 

TN tumors expressing androgen receptor have attracted the attention of the researchers in the past few years. They are identified as luminal androgen receptor (LAR) subtype because of the apocrine histological differentiation similar to luminal HR positive tumors. Compared to basal-like TN, LAR tumors are associated with lower grade, occur in postmenopausal women of older age, and have low pCR rate when treated with neoadjuvant chemotherapy [12]. The clinical presentation of this subtype is associated with a more frequent lymph node involvement at the time of initial diagnoses and a peculiar tropism with higher incidence of metastatic bone disease [13]. Interestingly, TN tumors with lobular histopathology display LAR features. LAR subtype is associated with high expression of genes involved in steroid hormone synthesis and androgen metabolism (*AR, FOXA1, GATA3*) and low expression of basal-like proliferation genes. 

Another intrinsic subtype identified as Claudine-low, was found to share similar features with basal-like subtypes. This subtype has high expression of stem cell-like and mesenchymal clusters and low expression of epithelial and intercellular tight-junction genes. Claudine-low tumors have metaplastic and/or medullary histological presentation and show relatively high frequency of lymphocytic infiltration. Their clinical significance is not yet well defined however research data are still under evaluation [14,15].

### 2.2. DNA Damage Repair in TNBC

#### 2.2.1. BRCA1 and BRCA2 Germline Mutations

In addition to the distinct molecular subtypes discussed. a very important component of TNBC in patients with this subtype is the *BRCA* alteration status. The prevalence of *BRCA* germline (*BRCAg*) mutations among patients with TNBC is about 10–30%. On the other hand, approximately 80% of breast cancers that present in patients with *BRCA1* germline mutations are TN tumors with basal-like profile. Mutations in the *BRCA2* gene are associated most likely with HR positive tumors and BC in males [16,17,18,19]. The lifetime risk of developing BC for women who inherit *BRCA* mutations is approximately 65% and 45% for *BRCA1* and *BRCA2* genes respectively [20]. Individuals who inherit mutations in *BRCA* genes are at higher risk of developing not only BC, but also high-grade serous ovarian carcinoma (HGSOC), pancreatic, gastric, and other solid tumors [21].

*BRCA1* and *BRCA2* are tumor suppressor genes that belong to the homologous recombination (HR) repair pathway. Human DNA is persistently in a dynamic self-control and self-repair state in order to maintain its integrity. Errors accumulated during replication or damage caused by exogenous agents trigger a cascade of signal events, known as DNA damage repair (DDR) mechanism. Failure of this mechanism to correct DNA lesion leads to uncontrolled cell proliferation, which constitutes the hallmark of tumorigenesis [22]. DDR machinery has at least five well-elucidated pathways that are respectively activated by specific type of DNA damage. Base excision repair (BER) is programmed to repair small, single strand base lesions while Nucleotide excision repair (NER) is responsible for larger, single strand, helix-distorting lesions. Mismatch repair (MMR) pathway acts on nucleotide defects responsible for double strand mismatches. Double strand breaks are detected and repaired by two overlapping pathways, HR and Non-Homologous End-Joining (NHEJ) [23,24].

HR pathway is considered the most important, high-fidelity DNA repair mechanism, responsible to correct DNA double strand breaks and maintain genomic stability. It acts during S and G2 phases by using as template for the damaged genetic information the normal sister chromatid [25]. This pathway is compounded by a large number of proteins that act in a rigorously synchronized manner. Ataxia telangiectasia-mutated (*ATM*) kinase is the upstream element that initiates the cascade of HR repair signaling and activates the subsequent effectors such as *BRCA1/2, PALB2, RAD51, BARD1, BRIP1, PARP1, TP53, CHK2*, and many other proteins involved in the pathway. Alterations on the genes encoding for the above proteins lead to HR repair deficiency (HRD) and increased risk of cancer [26]. The association of *BRCA1* and *BRCA2* germline mutations with DDR deficiency and tumorigenesis is very well described. In addition, somatic mutations of *BRCA1* and *BRCA2* genes or epigenetic alterations such as promoter methylation are also associated with impairment in DNA repair mechanism [27].

#### 2.2.2. *BRCA*ness Phenotype

The concept of *BRCA*ness was introduced more than ten years ago and describes the phenotypic similarities between *BRCA* germline-mutated and sporadic somatic *BRCA*-altered tumors [28]. Through the years, this notion has evolved by including a larger spectrum of genes involved in HR repair pathway. Therefore, *BRCA*ness is now defined as gene alterations leading to HR repair defects other than germline *BRCA* mutations [29]. Genetic, epigenetic or somatic alterations in *PALB2, ATM, RAD50, RAD51, CHEK2, FANCA, TP53, BRIP1*, and other genes of HR pathway were found to be associated with DDR deficiency and increased risk of cancer [30]. These alterations are less frequent in TN tumors compared to *BRCA1* and *BRCA2* gene mutations associated cancers. The detection of HR deficient tumors is very important because this cohort of patients may potentially benefit from DDR targeted agents, however, their efficacy outside the context of *BRCA1* and *BRCA2* mutations is still under investigation. Approximately 35% of TN tumors exhibit HR repair deficiency, which, make them particularly sensitive to drugs that act through DNA damaging such as platinum agents and PARP inhibitors (PARPi) [31].

Over the past few years, many studies have been focused on developing companion molecular tests that detect and quantify HRD, hence, predict tumor response to platinum agents and/or PARPi. To date, there is not yet been validated a standard HRD score that will help to identify patients who will benefit from HRD targeted treatment [32].

#### 2.2.3. PARP Activity

Poly ADP-ribose polymerase 1 and 2 (PARP1 and PARP2) proteins are key regulator enzymes that sensor DNA single-strand breaks (SSBs) and coordinate DNA repair response. They belong to base excision repair (BER) pathway and are important for the HR repair process. PARP1 and PARP2 catalyze the synthesis of branched poly ADP-ribose long chains, which in turn ribosylate and activate subsequent effectors of the DNA repair process. More specifically, PARP1 binds to the DNA damaged site, which generates allosteric changes in the PARP1 structure and initiate the enzymatic process [33]. Upon activation, poly ADP-ribosylation (PARylation) is catalytic for the recruitment of downstream DNA repair effectors such as DNA ligase III, DNA polymerase β, topoisomerases and XRCC1 and the chromatin structure remodeling [34]. After completing his mission, PARP1 ribosylates itself (autoPARylation) transforming in a negatively charged protein to facilitate the release from the repaired DNA site. In addition to the restoration of SSBs, PARP enzymes are also found to be very important for the HR repair process. There is evidence that PARP1 monitors the activity of *BRCA1* during DSB repair and prevent overreactions, which can also lead to genomic instability [35]. Apart from the DNA repair function, PARP1 and PARP2 are implicated in many other important processes such as transcription and immune function [36,37].

When replication fork comes upon DNA SSBs, it is very likely to collapse and generate DSBs, requiring the involvement of HR to repair them. HR deficient tumor cells such as *BRCA*- mutated ones, relies on the other DNA repair pathways and especially PARP1 and PARP2 enzymes activity in order to survive. Thus, inactivation of PARP in HR deficient tumors will result in cell death, which is known as “synthetic lethality” principle (Figure 1). The application of this principle in cancer treatment resulted to the development of PARP inhibitor drugs. In addition to synthetic lethality, a direct cytotoxic effect of PARPi has also been proposed. Some PARPi can act by preventing autoPARylation and “traping” PARP protein at the DNA site, a mechanism of action similar to Topoisomerase II inhibitors [38].

Over the past decade, a large number of clinical studies have been investigating the efficacy of PARPi and recently this category of drugs received approval for the treatment of *BRCA 1/2* mutated high-grade serous ovarian cancer (HGSOC) and TNBC [39,40,41,42].

HR deficient tumor cells such as *BRCA*- mutated ones, relies on the other DNA repair pathways and especially PARP enzymes activity in order to survive. Thus, inactivation of PARP in HR deficient tumors will result in cell death, which is known as “synthetic lethality” principle.

## 3. Targeted Pathways in TNBC

The advancement in sequencing technologies enabled the identification of important molecular pathways involved in tumor progression and resistance to treatment [43]. As our knowledge in regard to the underlying molecular biology of TNBC has made enormous progress, the treatment of TNBC is shifting towards biologically informed approaches. Tailored treatment strategies based on the expression of molecular biomarkers is an area of great excitement that is growing very rapidly. For TNBC, inter and intra-tumor heterogeneity make the translation of targeted treatments into meaningful clinical outcomes very challenging. Years of research efforts led to the development of PARPi which are the first targeted agents approved by the regulatory agencies for the treatment of *BRCA*-mutated tumors including TNBC. Immune checkpoint inhibitors have also displayed great activity in TN tumors. Other targeted agents are currently under evaluation with promising preliminary results. Below are underlined the most relevant molecular pathways in TNBC (Figure 2).

### 3.1. Targeting DNA Repair Pathway

#### 3.1.1. Platinum Salts

The role of platinum agents in the treatment of TNBC has been extensively studied over the past decade. These drugs act through cross-links DNA damage-mediated apoptosis of the tumor cells. A large number of studies, including large randomized clinical trials, have extensively evaluated the benefit from the addition of cisplatin or carboplatin to the standard anthracycline and taxane-based neoadjuvant chemotherapy (NACT) [44,45,46] (Table 1). The results showed that patients who received platinum-based chemotherapy had a higher response rate and were more likely to achieve pathological complete response (pCR). Remarkably, HRD was associated with higher sensitivity to platinum agents and two-fold increased pCR rate. Patients who acquired pCR after neoadjuvant treatment had better long-term outcomes compared to patients with residual disease. 

In accord the above, GeparSixto, a phase II clinical trial, investigated the addition of carboplatin to neoadjuvant regimen taxol, non-pegylated liposomal doxorubicin, and bevacizumab in patients with stage II–III TNBC. Patients who received carboplatin had higher chance to achieve pCR (53.2% vs. 36.9%, *p* = 0.005) [49]. Survival data reported last year showed better DFS (HR 0.56, *p* = 0.022) for patient treated with carboplatin, while there was no statistically significant improvement in OS [50]. In subset analysis, 70,5% of tumors displayed HRD and 60% of them had high HRD score in the absence of *BRCA* mutations. HRD was independent predictive factor of pCR (OR 2.6, *p* = 0.008) and the addition of carboplatin increased the pCR rate approximately 50% (63.5% vs. 33.9%, *p* = 0.001) in HR deficient tumors but had minimal effect on HR proficient ones (29.6% vs. 20.0%, *p* = 0.540). HR deficient non-BRCA-mutated tumors had also higher pCR rate when treated with carboplatin (63.2% vs. 31.7%, *p* = 0.005) [51]. A secondary analysis of this trial from Hahnen and colleagues reported that patients who harbored a deleterious germline *BRCA1* or *BRCA2* mutation had a superior response rate to chemotherapy, which was not improved by the addition of carboplatin. In contrast, patient without *BRCA1* or *BRCA2* mutation had the greater benefit from the addition of carboplatin [52]. 

Similarly, the CALGB 40603 trial evaluated 443 previously untreated patients with clinical stage II–III TNBC who received carboplatin and/or bevacizumab in addition to standard neoadjuvant chemotherapy. The pCR rate in the breast, which was the primary endpoint, was significantly improved with the addition of carboplatin (60% vs. 44%, *p* = 0.0018) or bevacizumab (59% vs. 48%, *p* = 0.0089) but only carboplatin showed a statistically significant increase in pCR in both breast and axilla (54% vs. 41%, *p* = 0.0029) [44].

Recently reported survival data from this trial, after a median follow-up 5.7 years, showed better event free survival (EFS) (86.4% vs. 57.5%) and OS (88.7% vs. 66.5%) in patients who achieved pCR. These results were similar in all treatment arms. For patients with residual disease, no long-term survival benefit was observed with the addition of carboplatin (HR 0.99) or bevacizumab (HR 0.91). Of note, this trial was not powered to detect long-term survival outcomes [53]. 

In an effort to enhance treatment effectiveness in TNBC, studies evaluated the combination of PARPi with platinum-based chemotherapeutic backbone based on the rationale that both agent act through DNA damage mechanism. The I-SPY-2 study evaluated 60 patients with TNBC who received 12 weeks paclitaxel followed by four cycles of dose dense doxorubicin and cyclophosphamide (ddAC) with or without carboplatin and PARPi. The pCR rate was 51% in the experimental arm compared to 26% in the standard treatment. Given the study design, it is not clear what the role of carboplatin in the outcomes [54]. According to BrighTNess study, a similar phase III trial with 634 treatment naïve TNBC patients that investigated the role of carboplatin and/or veliparib to the standard neoadjuvant regimen, the pCR rate was 58% and 53% in the arms treated with carboplatin compared with 31% in the control arm. In this trial the addition of veliparib was found to have no impact in pCR rate. In regard with *BRCA* mutation status, this trial did not show any difference in pCR rate between BRCA mutated and wild type tumors (47% vs. 48%). However, the HRD tumor status in the study population has not been reported [55]. In contradiction with the above results, Byrski et al, reported a 61% pCR rate in 107 *BRCA1* mutated, stage I to III breast cancer patient treated with four cycles of neoadjuvant cisplatin followed by surgery and adjuvant chemotherapy. This is a significant response considering that cisplatin was administrated as monotherapy [56].

#### 3.1.2. pCR as Surrogate Marker of Survival Benefit

Interestingly, the primary endpoint of most of the above described neoadjuvant studies, was the pCR rate. The importance of pCR and its strong correlation with long-term survival outcomes has been extensively studied. Two large meta-analysis, have shown that the pCR rate in TNBCs treated with platinum drugs is similar with that observed in HER2+ disease treated with trastuzumab and pertuzumab [57]. It is worth mentioning that in HER2+ disease, the administration of trastuzumab and pertuzumab is associated with significant improvement in event-free survival and overall survival [58,59]. However, in TNBC, according to these meta-analyses, the achievement of pCR is clearly associated with an improvement in DFS rates, but its impact in OS still under evaluation. An explanation for these results could be the discrepancy on the definition of pCR and whether it is achieved only in breast or in both breast and axilla. Specifically, the correlation with better long-term outcomes is stronger when pCR is determined as absence of in situ and invasive residual disease in both breast and axilla.

The first meta-analysis of seven neoadjuvant trials performed by Minckwitz et al, evaluated the association of pCR with survival outcomes in 6377 patients with BC. In this study, DFS was significantly higher in patients who had no residual disease either invasive or DCIS in both breast and axilla compared to those with any residual invasive or DCIS in breast or LNs (HR 0.446, *p* < 0.001). Patients with TN tumors who achieved pCR had very good prognosis [60]. A second, large pooled analysis of twelve neoadjuvant trials, the CTNeoBC study, evaluated the association of pCR with long survival outcomes in 11995 patients. pCR was defined as eradication of the invasive or in situ disease in breast and axilla. The results showed that pCR in breast and LNs had stronger correlation with EFS (HR 0.44) and OS (HR 0.36) compared to pCR in breast alone (HR 0.60 for EFS and 0.51 for OS). The association was stronger for TN tumors (HR 0.24 for EFS and 0.16 for OS). However, this study failed to validate pCR as marker for improvement of EFS and OS [61].

For the metastatic disease, platinum drugs were found to be effective in BRCA mutated patients. A randomized phase III trial, TNT study, evaluated 376 patients with metastatic TNBC or BRCA1/2 germline mutation who received docetaxel (100 mg/m^2^) or carboplatin (AUC 6) every three weeks for up to 8 cycles or until progression. Patients with a BRCA1/2 germline mutation who received carboplatin had significantly higher ORR compare to those who received docetaxel (68% vs. 38%, absolute difference 34.7%). PFS was also higher (6.8 vs. 4.4 months), but there was no difference in OS. Furthermore, non-basal-like TNBC probably showed a better response to docetaxel compared to carboplatin [62].

In summary, the addition of platinum agents to standard NACT significantly improved DFS in TNBC. Long-term survival analyses also support the use of this category of drugs in the neoadjuvant setting. pCR is associated with better long-term survival outcomes and residual disease may indicate the necessity for treatment escalation. Patient with BRCA1/2 mutations have better response to carboplatin compared to taxanes. The activity of platinum drugs seems not to be limited only to the triple negative *BRCA1* and *BRCA2* mutated tumors, but also to the sporadic TN tumors that harbor DDR deficiency.

#### 3.1.3. PARP Inhibitors

PARPi are the first approved targeted treatment and probably the most promising therapeutic strategy for TNBC. They possess antiproliferative and proapoptotic properties and act as chemosensitizers and radiosensitizers. PARPi seem to have synergistic effect with platinum agents, topoisomerase I inhibitors and radiotherapy [63,64]. The combination of PARPi with platinum agents seems to be synergistic not only because of the DNA damage effect of platinums but also by the activation of caspase-mediated apoptosis independently of p53 activation [64]. Coupling these drugs with radiation (IR) has been shown to enhance radiation effect because of the inhibition of DNA damage repair. In addition, it was suggested that IR inhibits HR repair pathway through cytoplasmic inactivation of *BRCA1* which coupled with PARPi induce synthetic lethality in otherwise HR proficient cells. This activity seems to be dependent on p53 activity [65].

PARPi differ from each other in terms of potency and cytotoxic effect, even though belong to the same drug category [66]. Several studies have investigated the benefit from PARPi as a single agent or in combination with chemotherapy in TNBC. Their activity in *BRCA1* and *BRCA2* mutated tumors is now well established while the impact on the sporadic TNBC is still under investigation (Table 2).

Olaparib was the first PARPi, which received approval on January 2018 for the treatment of *BRCA* mutated BC. OlympiAD trial, a phase III randomized study, investigated the efficacy of olaparib in patients with metastatic, HER2 negative BC who were carriers of deleterious germline *BRCA1* and *BRCA2* mutations. Patients were 2:1 randomly assigned to receive either single agent olaparib 300 mg daily or treatment of physician’s choice, which consisted in single agent capecitabine, vinorelbine or eribulin. PFS was the primary end point of this study and was higher in the olaparib arm compared to the standard chemotherapy (7.0 vs. 4.2 months, *p* < 0.001). This study also reported quality of life (QoL) data based on patient reported outcomes, which were better in the olaparib group [67]. The recently reported survival data from OlympiAD did not show a relevant difference in median OS between the two arms (19.3 vs. 17.1 months, *p* = 0.513). However, this trial was not powered to detect OS differences and the uncontrolled treatment crossover after discontinuation of the study drug might be a confounding factor in the statistical analysis. In sub-group analyses, there was a 7.9 months difference in the olaparib arm compared to standard chemotherapy arm for patients who had received no prior treatment in the metastatic setting (25.6 vs. 14.7 months, *p* = 0.02). This outcome needs to be validated in prospective studies with larger patient population [68]. Clinical trials are underway in order to determine and establish the activity of olaparib in early *BRCA* mutated BC. Talazoparib is a dual mechanism PARPi that inhibits PARP enzymes and traps it in the DNA, driving *BRCA* mutated tumor cells to apoptosis. This fact classifies it among the most potent and cytotoxic PARPi [69]. EMBARCA was the phase III randomized clinical trial that evaluated the efficacy of talazoparib as a single agent versus physician’s choice treatment in metastatic *BRCA*-mutated BC. This trial enrolled 431 patients who were randomized 2:1 to receive talazoparib 1mg/kg daily or chemotherapy with single agent capecitabine, vinorelbine, eribulin or gemcitabine and primary end point was median PFS. The results favored the talazoparib arm with median PFS 8.6 months compared to standard treatment 5.6 months (*p* < 0.0001). The ORR for patient who received talazoparib was 62.6% vs. 27.2% for those who received chemotherapy. Based on this trial, talazoparib received regulatory approval for the treatment of *BRCA*- mutated, HER2 negative metastatic BC [70].

Pooled analysis of the two studies confirmed the reported results for improved PFS with single agent PARPi compared to standard chemotherapy for *BRCA* mutated mBC patients. However, it is not clear the treatment sequence in rapport with platinum agents since platinum drugs were not allowed in the control arm and there is no head to head comparison between the two drug categories. Significant improvement was also noticed in QoL parameters (HR 0.40, 95%CI (0.29–0.54)) [71]. 

Veliparib is another drug of PARPi family and is considered less cytotoxic compared to other members. This can be explained by the fact that veliparib act through inhibition of PARylation but has no ability to trap PARP1 in the DNA [72]. A phase II randomized study, BROCADE trial, compared the activity of carboplatin and paclitaxel with either veliparib or placebo in *BRCA* mutated, metastatic or locally recurred BC patients. This study did not meet its primary end point, which was PFS, however patients who received veliparib had a higher response rate and greater chance to achieve complete response. A third arm with patients receiving the combination of veliparib and temozolomide, which was showed to be effective in preclinical models, was inferior compared to carboplatin and paclitaxel-based regimens [73]. 

Two other drugs, rucaparib and niraparib are being studied for their efficacy in patients with germline *BRCA* mutated BC. In the BRE09-146 trial of Hoosier Oncology Group, rucaparib failed to improve outcomes in 128 patients with TNBC or *BRCAg* mutations who received cisplatin with or without rucaparib for residual disease after anthracycline/taxane based treatment [74]. BRAVO trial, a phase III randomized study, is assessing the activity of niraparib compared to TPC in *BRCA* mutated, HER2 negative mBC patients and the results are awaited. (NCT01905592)

The combination of PARPi with platinum agents has been a very intriguing field of research in the past years. However, the addition of PARPi to platinum drugs seems to have no impact in clinical outcomes [73,74]. The rationale for administration of PARPi concurrently with radiation lie on the fact that radiation act through DNA damage and inhibition of PARP will prevent the DNA repair and eventually drive tumor cells to apoptosis [75]. Recently, special interest has been expressed in the combination of PARPi with immunotherapy or *AKT* inhibitors. Prospective studies will show the outcomes from these combinations. The use of PAPPi and combination strategies not only in the context of *BRCA* mutations, but also on the broader landscape of HRD tumors and the validation of biomarkers that will predict benefit from this category of drugs are the fields of future research efforts.

### 3.2. Antiangiogenic Agents

Bevacizumab is a recombinant humanized monoclonal antibody that blocks the vascular endothelial growth factor A (VEGF-A). The inhibition of VEGF-A impairs tumor neovasculature, which is necessary for tumor growth and tumor cell migration [76]. Multiple studies have been performed to investigate the role of Bevacizumab in TNBC. Two large randomized trials evaluated the combination of standard neoadjuvant chemotherapy and bevacizumab with or without carboplatin. These studies reported a 10% increase on the pCR rate, which is attributed to bevacizumab. On the other hand, treatment with bevacizumab was associated with higher rates of adverse events such as hypertension, neutropenia, mucositis, which led to dose reduction and postoperative complications [44,45,49,77]. Long-term outcomes that correspond to the impact of bevacizumab on pCR have not yet been clarified. Similarly, the GeparQuinto trial evaluated the addition of bevacizumab to standard NACT in 663 patients with TNBC. pCR rate was higher in the bevacizumb arm compared to control arm (39.3% vs. 27.9%, *p* = 0.003). Subset analysis for patients with *BRCAg* mutations showed significant higher pCR rate in patients with a *BRCA1* or *BRCA2* mutation who received bevacizumab compared to wild type patients treated with bevacizumab (61.5% vs. 35.6%, *p* = 0.004) [78]. 

In the adjuvant setting, BEATRICE study investigated the addition of bevacizumab to anthracycline and taxane-based chemotherapy did not show a statistically significant improvement in disease-free or overall survival however patients in the bevacizumab arm had a slight DFS benefit. A reasonable explanation for these results will be the heterogeneity of TN tumors and will be important to find the molecular subgroup, which significantly will benefit from the antiangiogenic drugs [79,80]. The combination of bevacizumab with paclitaxel for metastatic TNBC patients as a first or second-line treatment improves time to progression and response rates, but doesn’t have an impact on overall survival. A phase III randomized clinical trial evaluated the efficacy of paclitaxel (90 mg/m^2^ on D1,8,15) with or without bevacizumab (10 mg/kg on D1,15) every four weeks in 722 HER2 negative metastatic patients. PFS was higher in the combination arm (11.8 vs. 5.9 months) and ORR 36.9% vs. 21.2%, *p* < 0.001. However, there was no difference in median OS (26.7 vs. 25.2 months, *p* = 0.16) [81]. In conclusion, the role of bevacizumab in metastatic TNBC is well-established while for the treatment of early stage disease, its use is not encouraged outside the context of clinical trials. 

### 3.3. Immunotherapy 

Over the last years, immunotherapy has been one of the most promising and rapidly progressing areas of cancer therapy. Programmed cell death ligand 1 (PDL-1) is expressed in 20% of TNBC [82]. Ongoing clinical trials are evaluating the role of antiPD-1/PDL-1 immune checkpoint inhibitors (CPIs) in triple negative cancer. As shown in Table 3, pembrolizumab, nivolumab, atezolizumab and durvalumab are the most important drugs being used in these studies. 

KEYNOTE-119 was a phase III clinical trial that assessed the activity of single agent pembrolizumab in metastatic TNBC. In this study, patients were randomly assigned to receive either pembrolizumab monotherapy or standard chemotherapy according to physician’s choice (capecitabine, vinorelbine, gemcitabine or eribulin) with primary endpoint being OS. Study results reported very recently showed that pembrolizumab monotherapy was not superior to TPC, despite the initial results from phase Ib and II studies, KEYNOTE-012 and 086 respectively, which demonstrated some encouraging activity [83,84]. In an effort to improve the very modest activity of single agent CPIs in TNBC, investigators are now evaluating the role of a combinatory approach of CPIs and chemotherapy. The rationale behind this strategy is based on the fact that chemotherapy may trigger immune activity through immunogenic cell death and taxanes in particular are suggested to have an impact in the activation of toll-like receptor and modulate the activity of dendritic cells, which altogether potentiate the overall immune response [85]. Immune checkpoint blockades on the other hand, “unleashes the brakes” for T-cell activity, generating a robust immune response. KEYNOTE-355, a phase III randomized clinical trial compared either pembrolizumab or placebo in combination with chemotherapy (nab-paclitaxel, paclitaxel or gemcitabine plus carboplatin) as first line treatment in metastatic TNBC. Results from this trial are awaited. (NCT02819518).

Following the enthusiasm for the combinatory approach, CPIs and chemotherapy in advanced BC, KEYNOTE-173, a phase Ib multicohort study, looked at the activity of pembrolizumab in combination with different doses and schedules of taxane, carboplatin and anthracycline neoadjuvant chemotherapy in TNBC patients. Results reported at last SABCS demonstrated a 60% pCR rate and ORR reached 100% in nab-paclitaxel/carboplatin/pembrolizumab arms [86]. Subsequently, an ongoing phase III clinical trial, KEYNOTE-522, is investigating the efficacy of neoadjuvant treatment with the combination of either pembrolizumab or placebo and chemotherapy. Early data from recent interim analysis showed higher pCR rate in the pembrolizumab arm. Complete data analysis from this study is anticipated in the near future [87]. Another neoadjuvant study, I-SPY 2 phase II randomized clinical trial, compared the addition of pembrolizumab to standard paclitaxel/anthracycline chemotherapy with chemotherapy alone. The pCR rate in patients with TNBC was 60% in the pembrolizumab arm compared to 20% in the control arm. Of note, in this study chemotherapy regimen did not include a platinum drug, which explains the lower pCR rate in the control arm [88]. KEYNOTE-242, is an ongoing phase III randomized trial, is assessing the role of pembrolizumab as adjuvant treatment in TNBC patients with residual disease after neoadjuvant chemotherapy. (NCT02954874).

Atezolizumab is anti-PD-L1 CPI that inhibits the interaction of PD-1 with *B7-1*, which releases T-cell activity. It is the first FDA approved immune checkpoint inhibitor for the first line treatment of PD-L1 positive TNBC. IMpassion130 trial was the phase III randomized study that showed the clinical benefit from the addition of atezolizumab to nab-paclitaxel in patients with no prior treatment in the metastatic setting. IMpassion130 trial randomized 902 patients with TNBC to receive nab-paclitaxel in combination with either atezolizumab or placebo. In the intention-to treat population, median PFS was higher in the combination arm compared to nab-paclitaxel alone (7.2 vs. 5.5 months, *p* = 0.002), while in PD-L1 subgroup the difference was more profound (7.5 vs. 5.5 months, *p* < 0.001) [11]. The median OS for the general study population was not statistically significant between arms, 21.3 months in the atezolizumab arm and 17.6 months in the placebo arm (HR for death 0.84, *p* = 0.08). 

In the PD-L1 positive subgroup, formal statistical analysis for OS was not formally performed, however, there was a seven-month difference between atezolizumab and placebo arm, 25.0 vs. 18.0 months respectively [89]. Neoadjuvant and adjuvant studies are ongoing to investigate the activity of atezolizumab in early stage TNBC. 

Another very promising therapeutic approach seems to be the combination of CPIs with PARPi have displayed synergistic activity. The complex mechanism how PARP proteins and inhibitors interact with the immune system is not yet very well understood and sill an area of intensive research. PARP-1 was found to promote the activity of inflammatory cytokines such as IL-6 and TNFa and participate in the dendritic cell (DC) differentiation [90,91]. The inhibition of PARP will result in the attenuation of the immune response. However, it is associated with increase in TILs through CCL2/5 chemokines and upregulation of PDL-1 expression, which enhance the activity of CPIs [92,93]. Currently, a large number of clinical trials are investigating the efficacy of this therapeutic approach in *BRCA* mutated tumors and in the broader category of tumors that display HRD [94,95]. The results from these studies are anticipated with preliminary data displaying promising objective responses.

### 3.4. PI3K/AKT/mTOR Inhibitors

Alteration of the phosphoinositide-3 kinase (*PI3K*) signaling pathway has been related with angiogenesis, tumor proliferation and inhibition of apoptosis. Activating mutations of oncogenes (*PIK3CA*, *AKT*, *mTOR*) or inactivating mutations of tumor suppressor genes (*INPP4B*, *PTEN*) are associated with tumor growth and treatment resistance [96]. In hormone HR positive disease everolimus, an *m-TOR* inhibitor, is being used to overcome resistance to endocrine treatment and *PI3k* inhibitor alpelisib, showed activity in hormone resistant, *PIK3CA* mutated tumors [97,98].

In TNBC, *PIK3CA* is the second most frequently detected mutation after *TP53*, yet the implication of *PI3K/AKT/mTOR* inhibitors in this subset of BC is not well defined [99]. Several studies have investigated the impact of m*-TOR* inhibitors in TNBC. Everolimus was the drug that was used in these trials. The combination of everolimus with chemotherapy did not show any benefit in disease free and overall survival or any difference in pCR. However, this combination was well tolerated. Further investigation in the future might identify the most effective combinations or the subgroup of patients most likely to respond to m*TOR* inhibitors [45,100].

Pan-class I *PI3K* inhibitor Buparlisib failed to show any activity when given in combination with paclitaxel in HER2 negative metastatic BC patients and specifically in the TNBC subpopulation the combination was inferior compared to paclitaxel alone [101]. The results were similar in both intention-to-treat and *PI3K*-altered cohort which was defined as *PIK3CA* mutations or low *PTEN* expression. In contrast, *AKT* inhibitor ipatasertib in combination with paclitaxel showed higher PFS compared to paclitaxel in unselected TNBC patients with no prior treatment for the metastatic disease. In *PIK3CA/AKT* activated or *PTEN* inactivated patients the results were also favoring the combination arm. An ongoing randomized phase III trial is evaluating the combination of ipatasertib with paclitaxel in *PIK3CA/AKT/PTEN* altered TNBC [102].

Very promising preliminary data have been reported form the randomized, phase II study of *AKT* inhibitor capivasertib in combination with paclitaxel versus paclitaxel plus placebo in TNBC, PAKT trial. A cohort of 140 treatment naïve metastatic TNBC patients were 1:1 randomized to receive capivasertib 400 mg or placebo administered on specific days of a 28 days cycle (2–5, 9–12, 16–19), in combination with paclitaxel 90 mg/m^2^ weekly, D1,8,15 q 28 Patients who received capivasertib had a 5.9 months PFS compare to the placebo arm 4.2 months, median OS was 19.1 and 12.6 months respectively. 28 patients with genomic alterations on the *PIK3CA, AKT1* or *PTEN* genes had 9.3 months PFS with capivasertib/paclitaxel compared to 3.7 months with placebo/paclitaxel. A larger phase III trial, Capitell290, will validate these data [103].

Additional clinical trials are investigating several, new *PI3K/AKT/mTOR* inhibitors in combination with different doses and/or chemotherapy regimens and will elucidate the role of this drug family in the treatment of TNBC.

### 3.5. Targeting AR Pathway

As discussed above, LAR subtype shares similar features with luminal HR-positive disease. Consistent with luminal type behavior is the fact that in subset analyses of neoadjuvant studies LAR showed the lowest pCR rate compared to other TN-phenotypes [104,105]. Interestingly, despite the low pCR rate LAR subtype seems to be associated with better survival outcomes [106]. 

In preclinical studies this subtype displayed sensitivity to antiandrogen blockade which prompted the initiation of clinical trials investigating the efficacy of AR-targeted agents such as bicalutamide, abiraterone and enzalutamide. AR expression was determined as nuclear staining >10% by IHC. The clinical benefit rate at six months was 29% for enzalutamide and about 20% for bicalutamide and abiraterone [107,108,109]. The difference in results could be explained by a couple of reasons. Primarily, comparing the mechanism of action of the above antiandrogens, Enzalutamide is a more potent drug, inhibiting androgen binding, *AR* translocation to the nucleus, and the transcriptional process within the nucleus. Bicalutamide has a partial androgen agonist effect while abiraterone, a CYP17 inhibitor, is administrated in combination with steroids which might as well possesses androgen properties. In addition, patient population in the enzalutamide study had one or less previous treatment lines and only those with bone metastases were eligible to participate, while in the abiraterone study were enrolled heavily pretreated patients and variable visceral metastases. Of relevance, in the enzalutamide trial researchers applied a stratification based on the androgen-driven gene signature named Dx-signature. Patients classified as Dx-positive showed significantly higher survival outcomes compared to the Dx-negative ones [110]. Although some long-lasting responses from androgen-deprivation therapy has been reported, the outcomes live much to be desired when compared to luminal ER/PR positive disease [111]. 

Prospective studies are focused on combination strategies that have been successfully applied in ER/PR positive tumors and are now being assessed for LAR subtype. Phase II clinical trials are underway, evaluating CDK4/6 inhibitors palbociclib and ribociclib in combination with bicalutamide. (NCT02605486, NCT03090165) 

Interestingly, *PI3K-AKT* pathway was found to be activated in almost 40% of *AR* positive TNBCs. The TBCRC032 trial is evaluating the efficacy of the combination enzalutamide with *PI3K* inhibitor taselisib and is currently suspended due to tolerance issues. Coupling antiandrogens with chemotherapy is also a very attractive approach that is being investigated for LAR tumors. An ongoing phase III trial is investigating the efficacy of enzalutamide monotherapy compared with the combination of paclitaxel with either enzalutamide or placebo. Several other trials investigating new antiandrogens or combination regimens are underway and their results are awaited to validate these new treatment approaches for LAR subtype. Furthermore, validation of the Dx-androgen-driven genomic signature is important in order to identify patient who will benefit from AR-targeted treatments.

## 4. Emerging Treatment Strategies

### 4.1. Antibody-Drug Conjugates

Antibody–drug conjugates (ADCs) are a newly developed sophisticated strategy for the cancer treatment. The technology behind these drugs is quite intriguing. In principle, ADCs are monoclonal antibodies linked to cytotoxic agents also known as “cytotoxic payload” through a cleavable or noncleavable linker. Cytotoxic payload usually is a very toxic agent which cannot be administrated alone due to unacceptable toxicity but when attached to an antibody is delivered directly inside the tumor cells. Antibodies are targeted against a tumor cell surface antigen and serve as a vehicle for the cytotoxic payload. They possess high affinity for the selected tumor antigen and upon binding to their target, lead to internalization of the ADC into the tumor cell [112,113]. Ado-trastuzumab emtansine is an ADC approved for the treatment of Her2new positive BC [114].

One of the most effective ADCs for TNBC is sacituzumab govitecan. It consists of a humanized IgG antibody that target tumor cell surface antigen trophoblast 2 (Trop-2) attached to the irinotecan metabolite SN-38 through a cleavable linker [115]. Trop-2 antigen is expressed in the majority of TNBCs. 

A phase I/II study evaluated the activity of sacituzumab govitecan in a cohort of 108 TNBC patients who had progressed in at least two prior treatment lines. Drug was administered intravenously, 10 mg/kg on days 1 and 8 every 21 days. The ORR was 33.3% including three complete and 33 partial responses. PFS and OS were 5.5 and 13.0 months respectively. Anemia and neutropenia were the most common observed side effects with Grade 3–4 toxicity occurring in >10% of the study patients [116].

Phase III randomized study is underway, ASCENT trial, which is investigating sacituzumab compared to physician’s choice eribulin, gemcitabine, capecitabine or vinorelbine. The study has completed accrual, 529 patients are enrolled and is expected to be reported in 2020 (NCT02574455).

### 4.2. Chemokine Receptors Pathway

Chemokines (CXC) are proinflammatory molecules involved in the innate immune response. In particular, CXCL8 is secreted mainly from blood monocytes but also from endothelial and epithelial cells. Its effect is mediated through CXCR1 and CXCR2 receptors which are G-protein-coupled cell membrane receptors overexpressed in endothelial and cancer cells but also in tumor associated macrophages [117]. Their role in tumorigenesis has been extensively studied for the past decades. Activation of CXCL8 signaling has been reported to increase cell proliferation and promote angiogenesis, tumor invasion, and metastases. In addition, preclinical studies have demonstrated that CXCL8 is involved in the regulation of breast cancer stem-like cells which are associated with tumor initiation, growth and dissemination [118,119]. 

Targeting *CXC* signaling pathway seems to be a very promising treatment approach for cancer treatment including BC. Several CXCR1/*2* receptor inhibitors are being tested in preclinical BC models and early phase clinical trials. Reparixin is a CXCR1/2 inhibitor that was tested in phase Ib study in combination with paclitaxel in in patients with Her2new negative metastatic BC. According to this study combination treatment of weekly paclitaxel on day 1,8 and 15 every 28 days cycle with oral reparixin 1200 mg three times daily for 21 days every 28 was well tolerated. The response rate was approximately 30% with two patients displaying >12 months durable response [120].

A phase II double-blind randomized study, FRIDA trial is ongoing and is investigating the efficacy of paclitaxel with either reparixin or placebo in metastatic TNBC patients (NCT02370238).

## 5. Evolving Molecular Landscape in TNBC

Despite our growing understanding of the underlying tumor biology, treatment of TNBC remains a challenge. Intra tumor heterogeneity (ITH) and clonal evolution are strongly correlated with de novo or acquired treatment resistance in this subset of BC. The numerous neoadjuvant studies have given tremendous insight on the prognostic significance of residual disease, tumor molecular evolution and possible mechanisms of resistance to treatment. Balko et al. performed molecular profiling of the residual disease after neoadjuvant treatment in 111 TNBCs. NGS analysis of 74 tumors revealed the most frequent alterations on the residual tumors were TP53, amplification of *MCL1* antiapoptotic gene, *MYC* amplification, *PTEN* deletion, *JAK2* amplification, as well as *CCND1* and *CDK6* amplification. NGS analysis of 20 paired pre and post-treatment biopsies showed enrichment in *PI3K/mTOR* pathway, regulators of cell cycle and DNA repair genes. However, alterations of prognostic significance were *JAK2* amplification and *BRCA1* mutations associated with poor survival outcomes while *PTEN* deletions had favorable impact [121]. Single cell analysis is a more sensitive method on detecting ITH and clonal evolution leading to treatment resistance and progression of the disease. Kim and colleagues performed exome sequencing in paired frozen biopsies obtained from 20 TNBC patients before neoadjuvant treatment, after the second cycle and surgical specimen after completing 6 cycles NAC. Single cell analysis was conducted in eight patients. This study showed that genomic alterations found in the residual tumors were pre-existing in the initial sample and significantly expanded as adaptive response to chemotherapy [122]. Consistent with the above results, clonal expansion from the primary site was also demonstrated in the metastatic setting. DNA whole exome and RNA sequencing was performed in 16 primary tumors and 2–7 matched metastases for each patient in addition to a matched control normal tissue sample. Among primary tumors, six were treatment naïve, five had prior NAC, and five received NAC and radiation. In this study, it was found that, particularly in basal-like TN tumors, clonal heterogeneity arises from the primary tumor and is maintained and expanded in distant metastases. Furthermore, the majority of genetic drivers were copy number alterations [123]. All these data, taken together, emphasize again the extensive ITH to begin with in TNBC, responsible for treatment resistance and the value of molecular profiling to select targeted treatments.

### 5.1. Monitoring Treatment Response and Resistance

Repeat tissue biopsy is a common practice used by physicians in order to assess tumor recurrence or progression to treatment. Reevaluation of hormone receptors expression and HER2 status is sometimes not unusual to differ from the primary tumor, and is taken into consideration to guide treatment decisions. In addition, molecular profiling of the new tumor tissue can show preexisting or acquired genomic alterations that led to treatment resistance and identify new actionable targets. Tissue biopsy, however, remains an invasive procedure and in some cases is not feasible to perform it. 

Efforts made to create an easy method that will detect and monitor tumor evolution led to development of liquid biopsy. This method consists in the analysis of cell free DNA (cfDNA), circulating tumor cells (CTCs) and other genetic materials released from the tumor cells in blood, urine, cerebrospinal fluid or bone marrow. Liquid biopsy is a minimally invasive approach that has enable the real time monitoring of clonal evolution and treatment resistance [124,125]. Liquid biopsy can give valuable information on the volume of microscopic disease and genomic alterations that drive tumor progression and/or treatment resistance. These data serve as the foundation for selection of targeted treatments. 

The combination of cfDNA and tumor tissue molecular profiling maximizes the potential to identify actionable mutations corresponding to targeted agents for cancer patient. In addition to initial treatment decisions, repeating liquid biopsy at every tumor progressing time will give the necessary information on the mechanism of resistance and detect new mutations that can be used as subsequent treatment options. Researchers are now focusing on validating liquid biopsy as method to detect early disease or recurrence and guide treatment decisions based on the microscopic disease. 

There are limitations in the current genomic assays with the more important being difficulty distinguishing between driver and coexisting mutations. The future research in this area will expectedly address these issues.

### 5.2. Translation of Predictive Biomarkers 

Implementation of biologically informed treatments is the foundational principle of precision medicine. Genomic alterations on tumor cells identified either in tissue or blood samples, are used as treatment targets. In TNBC the lack of the classic biomarkers such as HR and HER2, make the molecular profiling a very important and necessary tool to identify potential actionable genomic alterations. *BRCA* mutations and other DNA repair defects are predictors of response to PARP inhibitors. In the same way, any pathogenic genomic alteration that has a corresponding targeted agent should be considered a potential treatment option. A combination of different drugs might be necessary to achieve better results.

The evolution of immunotherapy on the other hand was the “game changer” in cancer treatment and its use is increasing very rapidly. Immune checkpoint inhibitors have shown excellent long-term survival outcomes in a large number of patients and different type of tumors. Thus far, there are limitations on the biomarkers being used to predict treatment response. PDL-1 expression, DNA mismatch repair instability and increased tumor mutational burden (TMB) are the biomarkers currently used to indicate patient that might benefit from immunotherapy [126,127,128]. However, not all patients who test positive for these biomarkers will respond to treatment and patient with low PD-L1 expression have shown clinical benefit [129]. A possible explanation for this fact will be the subsequent increase of PD-L1 expression from the immunogenic effect of treatment. Another parameter, tumor-infiltrating lymphocytes (TILs), was found to be associated with increased response to immunotherapy and better outcomes [130].

Also, increased TILs in residual tumors after NAC in TNBC patients were associated with favorable outcomes [131]. In addition, high absolute neutrophil counts and neutrophil-to-lymphocyte ratio before starting immunotherapy is associated with poor prognoses while isolated neutrophilia is not predictive of clinical outcomes [132,133,134]. The development of predictive tools to better select patients that will benefit from immunotherapy is a priority. Moreover, future efforts should be focused on discovering drug combinations which will increase tumor immunogenicity and enhance efficacy of immune checkpoint inhibitors. The optimal outcome from immunotherapy is to eventually develop a robust tumor-specific T cell memory that will continue to eradicate tumor cells for life-long.

## 6. Conclusions

In conclusion, TNBC remains a very challenging disease with poor prognoses compare to other subtypes of BC. New treatment strategies are warranted to improve the outcomes. The application of the knowledge we possess regarding the molecular biology of TNBC should be the cornerstone of the research efforts toward identifying novel therapies and other more effective drug combinations.

Personalized medicine seems to have a great significance for this very heterogeneous subset of BC. Tumor molecular profiling should be performed at diagnosis and after every tumor recurrence or progression and taken in consideration when to decide treatment plan. The simultaneous analysis of cfDNA will augment the chances to detect potential treatment targets.

## Figures and Tables

**Figure 1 cancers-12-00916-f001:**
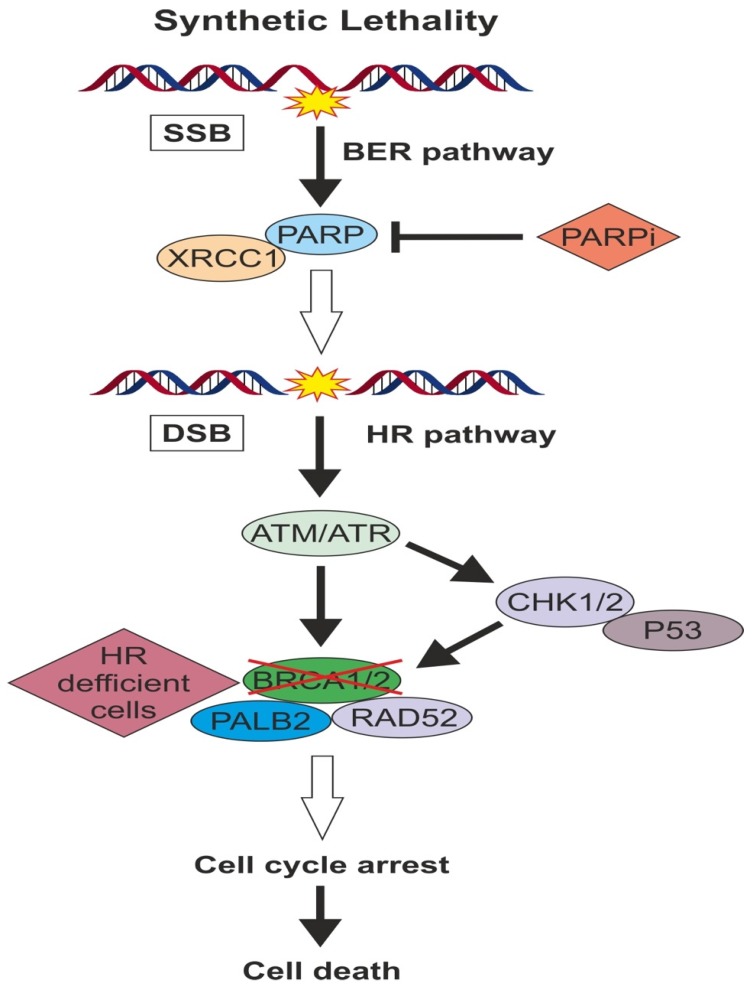
When replication fork comes upon DNA SSBs, it is very likely to collapse and generate DSBs requiring the involvement of HR pathway to repair them.

**Figure 2 cancers-12-00916-f002:**
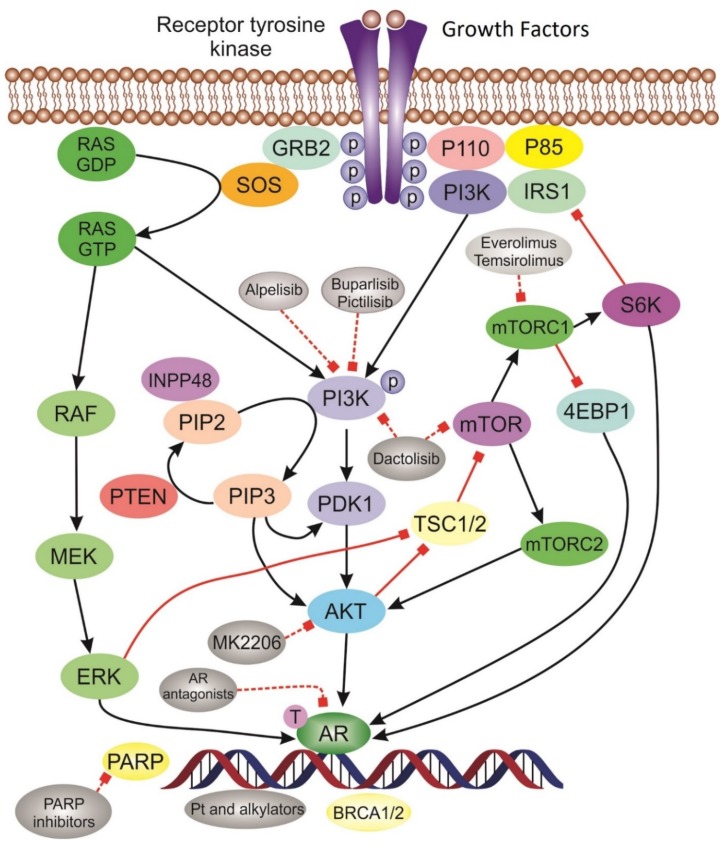
Schematic of the most relevant molecular pathways and targeted agents in TNBC. Tyrosine kinase receptor is activated upon binding of growth factors leading to activation of signaling pathways. *PI3K/AKT* pathway and inhibitor drugs: Phosphorylated PI3K activate AKT. The activation of AKT triggers downstream protein complexes mTORC activation that initiate gene transcription and promote cell growth. *AR* pathway is activated in LAR subtype tumors. Platinum drugs act through DNA damaging mechanism. PARP inhibitors induce “synthetic lethality” in *BRCA* deficient tumors.

**Table 1 cancers-12-00916-t001:** Results of platinum agents in the NA clinical trials.

Trial	Design	Phase	Population	Primary Endpoint	Results	Clinicaltrials.gov Identifier
CALGB40603	wPaclitaxel±Carboplatin±Bevacizumab Followed by dose dense AC vs. Standard NAC	II	Neoadjuvant Locally advanced TNBC	pCR	62.4% vs. 22.3%	NCT00861705
GeparSixto	Carboplatin+Bevacizumab+standard NAC vs. Bevacizumab+standard NAC	II	Neoadjuvant TNBC or HER2(+)	pCR	53.2% vs. 36.9%	NCT01426880
Jovanovic et al. [47]	wCisplatin+wPaclitaxel+Everolimus vs. wCisplatin+wPaclitaxel+placebo	II	Neoadjuvant Stage II/III TNBC	pCR	36% vs. 48%	NCT00930930
Zhang et al. [48]	Paclitaxel+Carboplatin vs. Paclitaxel+Epirubicin	II	Neoadjuvant Stage II/III TNBC	pCR	38.6% vs. 14.0%	NCT01276769
SHPD001	Cisplatin (4C)+wPaclitaxel (16 weeks)	II	Neoadjuvant LABC, including TNBC	pCR	64.7% in TNBC	NCT02199418
PreECOG 0105	Gemcitabine+Carboplatin+Iniparib	II	Neoadjuvant TNBC or BRCA1/2mt	pCR	62.4% vs. 22.3%	NCT00813956

Abbreviations: doxorubicin + cyclophosphamide (AC), pathological complete response (pCR), Neoadjuvant chemotherapy (NAC).

**Table 2 cancers-12-00916-t002:** Results of PARP Inhibitors in clinical trials.

Trial	Design	Phase	Population	Primary Endpoint	Results	Clinicaltrials.gov Identifier
OlympiAD	Olaparib vs. PCT	Phase III	Advanced/Metastatic gBRCA, ≤2 prior lines	PFS	7.0 vs. 4.2 months HR 0.58, *p* < 0.001	NCT02000622
OlympiA	Olaparib vs. placebo	Phase III	Early-stage gBRCA, adjuvant therapy	Invasive dicease free survival (IDFS)	Ongoing	NCT02032823
BROCADE 3	C + P + veliparib vs. C + P + placebo vs. Temozolamide + Veliparib	Phase II	Metastatic gBRCA, ≤0–2 prior lines lines	PFS	14.1 vs. 12.3 months HR 0.789, *p* = 0.227	NCT01506609
BrighTNess	C + P + veliparib → AC vs. C + P + placebo → AC vs. Placebo + placebo + P → AC	Phase III	Stage II or III TNBC Neoadjuvant	pCR	58% vs. 53% vs. 31% *p* < 0.0001	NCT02032277
I-SPY 2	C + P + veliparib → AC vs. C + P + placebo → AC	Phase II	Stage II or III TNBC Neoadjuvant	pCR	51% vs. 26%	NCT01042379
EMBRACA	Talazoparib vs. PCT	Phase III	Advanced/Metastatic gBRCA, ≤3 prior lines	PFS	8.6 vs. 5.6 months	NCT01945775
BRAVO	Niraparib vs. PCT	Phase III	Advanced/Metastatic gBRCA, ≤2 prior lines	PFS	Completed accrual	NCT01905592

Abbreviations: doxorubicin + cyclophosphamide (AC), carboplatin (C), paclitaxel (P), germline BRCA (gBRCA); physician’s choice chemotherapy (PCT), progression free survival (PFS), pathological complete response (pCR).

**Table 3 cancers-12-00916-t003:** Summary of Clinical Trials for Immune Checkpoint Inhibitors.

Trial	Design	Phase	Population	Primary Endpoint	Results	Clinicaltrials.gov Identifier
Keynote-119	Pembro vs. TPC	Phase III	Metastatic TNBC	OS	Negative	NCT02555657
Keynote-355	Pembro + chemo vs. placebo + chemo	Phase III	Metastatic TNBC 1st line	OS	Ongoing	NCT02819518
Keynote-522	Pembro or placebo + P + C × 4 followed of Pembro or placebo + AC × 4 followed of Adjuvant Pembro or placebo	Phase III	Stage II–III TNBC Neoadjuvant/Adjuvant	pCR	Interim analysis favor Pembro arm	NCT03036488
Keynote-242	Pembro vs. observation	Phase III	TNBC patients with residual disease after NACT	IDFS in ITT and PD-L1 positive	Ongoing	NCT02954874
Impassion-130	Atezo + Nab-Paclitaxel vs. placebo + Nab-Paclitaxel	Phase III	Metastatic TNBC 1st line	PFS	7.2 vs. 5.5 months *p* < 0.001	NCT02425891
Impassion-030	adj T + AC / EC vs. Atezo + T + AC / EC followed of Atezo maintenance for 1 year.	Phase III	Stage II–III TNBC Adjuvant	IDFS	Ongoing	NCT03498716
Impassion-031	Atezo + Nab-Paclitaxel + AC vs. placebo + Nab-Paclitaxel + AC followed by adjuvant Atezo	Phase III	Stage II–III TNBC adjuvant	pCR	Ongoing	NCT03197935
ISPY-2	Pembro + chemo vs. placebo + chemo	Phase III	Stage II–III TNBC Neoadjuvant	pCR	60% vs. 20%	NCT01042379

Abbreviations: doxorubicin + cyclophosphamide (AC), carboplatin (C), paclitaxel (P), progression free survival (PFS), pathological complete response (pCR), Invasive disease-free survival, Atezolizumab (Atezo), Pembrolizumab (Pembro), Intent to Treat (ITT).

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
