# Peer review of "The Landscape of Targeted Therapies in TNBC"

_cancers, 2020, doi:10.3390/cancers12040916_

Round 1

Reviewer 1 Report

See Attached.

Author Response

Dear Reviewer, 

Thank you very much for your valuable comments. Please see attachment for the detailed responses. 

Sincerely,

Elena Vagia

Reviewer 2 Report

Congratulations to this comprehensive overview on the current landscape of TNBC and future directions of research which has been well elaborated on the basis of numerous references of the international literature. This paper clearly merits publication, however a list of typing errors has to be amended beforehand. Where stated, please add the literature recommended.

With kind regards

Typing errors / amendments / additional references - 

The respective lines of the manuscript are indicated: 

17: clonal evolution (Comment: Please quote Nik-Zainal S. et al.; Landscape of somatic mutations in 560 breast cancer whole-genome sequences; Nature 2016 Jun 2;534(7605):47-54. Here it is stated that mutations are acquired at earliest mutational state in TNBC.)

48: compared instead of compare

55: empty spaces

82: in view instead of in viwe

177: development instead of d evelopment

219: no benefit in OS (Comment: Please quote Hahnen E et al.; Germline Mutation Status, Pathological Complete Response, and Disease-Free Survival in Triple-Negative Breast Cancer: Secondary Analysis of the GeparSixto Randomized Clinical Trial; JAMA Oncol. 2017 Oct 1;3(10):1378-1385. In this paper it is stated, that the non-BRCA-mutated patients had the most benefit from the addition of Carboplatin.)

235: no benefit by Carboplatin or Bevacizumab (Comment: This applied when no pCR occurred.) 

305: mediated instead of madiated

340 : 27,2 % instead of 27,2

392: slight instead of slide

393: delete empty spaces

475: alpelisib instead of alpelicib

542: eliminate empty spaces

562: empty space is missing

569: endothelial instead of engothelial

581: patients instead of parients

588: numerous instead of noumerous

657: empty space is missing

677: when to deinstead of when

Author Response

Dear Reviewer, 

Thank you very much for your valuable comments. Please see the attachment with respective responses. 

Sincerely,

Elena Vagia
